# Eye Movement Desensitization and Reprocessing (EMDR) as a Possible Evidence-Based Rehabilitation Treatment Option for a Patient with ADHD and History of Adverse Childhood Experiences: A Case Report Study

**DOI:** 10.3390/jpm13020200

**Published:** 2023-01-23

**Authors:** Clotilde Guidetti, Patrizia Brogna, Daniela Pia Rosaria Chieffo, Ida Turrini, Valentina Arcangeli, Azzurra Rausa, Maddalena Bianchetti, Elisa Rolleri, Chiara Santomassimo, Gianluigi Di Cesare, Giuseppe Ducci, Domenico M. Romeo, Claudia Brogna

**Affiliations:** 1Pediatric Neurology, Università Cattolica del Sacro Cuore, 00168 Rome, Italy; 2Complex Operative Unit Prevention and Early Interventions (PIPSM), Department of Mental Health ASL ROMA 1, 00193 Rome, Italy; 3Clinical Psychology Unit, Fondazione Policlinico Universitario “A. Gemelli”, IRCCS, 00168 Rome, Italy; 4Department Women Children and Public Health, Catholic University of Sacred Heart, 00168 Rome, Italy; 5Pediatric Neurology Unit, Fondazione Policlinico Universitario “A. Gemelli”, IRCCS, 00168 Rome, Italy

**Keywords:** EMDR, adverse childhood experiences (ACE), ADHD

## Abstract

Background: Children with Attention Deficit Hyperactivity Disorder (ADHD) having a history of adverse childhood experiences (ACEs) could be very difficult to treat with standard psychotherapeutic approaches. Some children diagnosed with ADHD may have Post-Traumatic Stress Disorder (PTSD) or have had experienced a significant traumatic event. Trauma and PTSD could exacerbate ADHD core symptoms and be a risk factor of poor outcome response. Objective: to report for the first time the history of a patient with ADHD and ACE successfully treated with an EMDR approach. Conclusion: EMDR could be a promising treatment for ADHD children with a history of traumatic experiences in addition to pharmacological treatments.

## 1. Introduction

Adverse childhood experiences (ACEs) are the umbrella term used to explain traumatic experiences during childhood leading to a significant effect on the biological, psychological and social functioning on affected children. These experiences are usually caused by persons with a care role in the child’s life and can be direct (sexual abuse, psychological abuse, physical abuse, and neglect) or indirect (psychiatric illness, deaths, alcoholism or drug addiction of a parent), [1]. 

According to the American Psychiatric Association (APA) [2], trauma is described as a perceived experience that threatens injury, death, or physical integrity and causes feelings of fear, terror, and helplessness. These experiences may occur during a single event (acute) or as a result of repeated (chronic) exposure. Traumatic events include abuse, violence, neglect, loss, accidents, disasters, war, and other emotionally harmful experiences. Early childhood trauma, especially complex trauma, can cause neurobiological changes that impact human development and cause significant changes in brain functions trough loss of neurons and hormonal alterations [3]. These changes in brain structures could be responsible for cognitive and physical functioning. ACEs and their related disorders have been linked to structural and functional alterations of brain regions and circuits associated with regulation of emotion, stress and cognitive functions (e.g., anterior cingulate cortex (ACC), amygdala, hippocampus, limbic–hypothalamus–pituitary–adrenal axis (LHPA)) and the immune system [4,5]. 

In the last few years there has been a growing interest in understanding the possible relationship between hyperactivity-attention deficit disorder (ADHD) and ACEs [6,7,8,9,10,11]. It is known that adversity-related toxic stress may impact brain development, including regions implicated in ADHD symptomatology [6,7,8,9,10,11]. The most consistent neurobiological and behavioral findings in children exposed to adverse psychosocial experiences seem to be linked to impaired prefrontal cortex functioning. It has been suggested that children exposed to adverse childhood experiences could present abnormalities in prefrontal cortex functioning.

The negative long-term consequences of ACEs on physical and mental health are well-documented [8], with a substantial body of literature linking ACEs to an increased likelihood of developing ADHD [4,5,6,7,8]. Interestingly, a longitudinal study found that experiencing ACEs before age 5 years was associated with ADHD at age 9 years [9]. Childhood trauma could also potentially affect symptoms through memory processes and may be a risk for the persistence of ADHD into adulthood [10]. 

The relationship between ADHD and post-traumatic stress disorder (PTSD) has been of interest in the last years. PTSD is defined according to APA criteria as a complex clinical condition characterized by the presence of different clinical signs including disturbing thoughts, feelings or dreams related to the traumatic events, avoidance and trauma-related hyperarousal symptoms after being faced with trauma directly or indirectly, including a real or intimidating manner to death, severe injuries or sexual assault [2]. Epidemiological data revealed that PTSD represents the most frequently psychiatric disorder related to trauma. Most often PTSD and ADHD could be confused as there is a high degree of overlapping symptoms that could be shared. The comorbidity between these two conditions is well reported in the literature ranging from 12% and 37% across the life span even if not always reported [12,13,14]. 

Children with PTSD have also been shown to have a neuropsychological profile suggestive of impaired prefrontal cortex functioning including deficits in sustained attention, impulsivity and poor performance in cognitive measures evaluating abstract reasoning/executive function [15]. It is unclear whether impairment in prefrontal functioning in children with ACEs should be attributed to delayed maturation of the pre-frontal cortex, similar to findings in children with attention deficit and hyperactivity disorder [16,17] or to neuronal damage due to chronic stress. 

Several studies have shown high rates of ADHD [6,7,8,9,10,11] among children with ACEs. A recent longitudinal study made by Candelas and colleagues suggested that the reverse association between a childhood diagnosis of ADHD and occurrence of ACEs is also possible. Another study also showed that children with ADHD have higher ACE exposure compared with children without ADHD [18]. These findings suggest that ADHD may have a role in perpetuating the cycle of childhood adversity and highlight the need for treatment [11]. 

We report a case study of a patient with ADHD and history of complex childhood trauma treated using eye movement desensitization and reprocessing (EMDR) to target traumatic experiences. The study was conducted according to the guidelines of the Declaration of Helsinki, and approved by the EC of Fondazione Policlinico Gemelli (protocol number 0039538/21, date of approval 10/11/2021).

## 2. Case Presentation

A child of Brazilian origin was institutionalized in an orphanage at the age of 3 years and adopted by an Italian family at the age of 9 years together with one of his three sisters. The child’s developmental milestones were reported as normal. His biological mother suffered from substance abuse disorder (narcotic) due to which she was stripped of her parental authority. She died by an overdose in the bedroom of the house when he was 3 years old. It was also reported that the child witnessed the death of 2 sisters by a car accident when he was younger than 9 years. At the age of 12 years, he was treated on an outpatient basis at Service of Child Neuropsychiatry of our hospital where a diagnosis of ADHD, as defined by DSM-5 [2], was made. He also presented with some clinical signs resembling a mild presentation of PTSD. He was started on cognitive behavioral psychotherapy intervention (CBT) once a week with evidence of effectiveness. A neuropsychological assessment was performed at 12 years old using the Wechsler Intelligence Scale for Children 4th Edition (WISC-IV) which revealed an intellectual functioning in the normal range: QIT value of 121 (Verbal Comprehension Index (VCI) 124; Perceptual Reasoning Index (PRI) 132; Processing Speed Index (PSI) 115; Working memory index (WMI) 91). His social and academic functioning was mildly impaired. At the age of 13 years, he started modified release methylphenidate therapy (30 mg/day) in addition to the CBT, showing an improvement on hyperactivity and dis-attention symptoms. In the following two years, he had a traumatic experience related to the removal of his only sister who presented psychiatric problems. 

For this reason, he started to have symptoms including anger, irritability, loss of control, affective instability and oppositional-defiant attitudes. Therefore, he started aripiprazole therapy (5 mg increased to 10 mg a day) associated to modified release methylphenidate therapy with slight improvement of the affective symptoms and reduction of the irritability. He repeated neuropsychological assessment including the WISC-IV which revealed an intellectual functioning in the normal range (QIT value of 107, VCI 104, PRI 112, PSI 103, WMI 85) associated with severely specific learning difficulties. Successively, the patient refused to continue CBT therapy. Due to the persistence of oppositional-defiant attitudes and emotional difficulties, in agreement with the adoptive parent and the patient, it was decided to change the psychotherapy treatment plan from CBT to EMDR treatment considering also his previous traumatic experience. EMDR treatment was considered also due to the collaborative efforts between our hospital and the territorial Presidium Prevention of Early Interventions Mental Health (PIPSM). Before starting EMDR treatment he performed a clinical assessment evaluation with the following self-report questionnaires (Table 1): Symptom Checklist-90 Revised (SCL90-R) for the evaluation of symptoms and clinical severity; Personality Inventory for DSM 5 (PID-5) (Child Age 11–17) for measuring mental functioning; Childhood Trauma Questionnaire (CTQ) and Adverse Childhood Experience (ACE Questionnaire–ACEQ) for measuring traumatic and aversive childhood experiences; Difficulty in Emotion Regulation Scale (DERS) to assess emotional regulation and the degree of dysregulation present; Toronto Alexithymia Scale (TAS-20) for measuring the ability to recognize and discriminate their emotional experiences; Dissociative Experiences Scale (DES-II) to assess and to evaluate dissociative phenomena. He also performed some neuropsychological assessments (Table 2).

Following clinical assessment and consolidation of the therapeutic relationship, EMDR treatment was carried out. 

We used the standard Protocol of EMDR therapy on past memories. EMDR therapy consists of eight phases. These phases occur over multiple sessions, with one session sometimes using parts of several phases. An example of this would be how Phases 1 and 2 typically happen only in early sessions, while Phases 3 through 8 are part of multiple sessions later. For a single disturbing event or memory, it usually takes between three and six sessions. More complex or longer-term traumas may take 8 to 12 sessions (or sometimes more). Sessions usually last between an hour and 90 min (EMDR method is reported in Appendix A). 

The EMDR procedure was explained and we obtained consent for the treatment. Due to the consideration of the traumatic memories related to his mothers’ death (when his mother died, he was 3 years old and he remembered only a song sent by a radio: a very sad song), he was treated on this target with EMDR for two 65 min sessions. 

An avoidant atattachment disorder was considered during EMDR therapy. Even if the attachment disorder may mimic ADHD, in our case both behavioral and neuropsychological findings were compatible with ADHD according to DSM-5 criteria, but the severity was influenced by the attachment disorder and the related PTSD. 

A follow-up cognitive evaluation was administrated after 9 months of EMDR treatment on n.1 target (Table 2) showing an improvement on executive functions. In addition, the symptoms related to the emotional dysregulation had gradually diminished. 

## 3. Discussion

The symptoms of ADHD and trauma-related symptoms in growing children have certain similarities. In both there is difficulty in sustaining attention, an incapacity to listen, failure to complete duties, difficulty in organization, memory dysfunction, irritability, and restlessness. Trauma and PTSD could also exacerbate dis-attention, impulse regulation, and physiological hyperreactivity symptoms or disruptive behaviors with oppositionality and aggression [19]. Hyperactivity in children with PTSD could be a result of increased muscular tension because of anxiety. 

Nowadays, ADHD diagnosis includes a wide variety of clinical pictures, reflecting the complexity of his etiology. In clinical practice it often occurs that symptoms that have developed as a result of trauma, single or repeated, can be mistakenly confused with the diagnosis of ADHD. Certainly, further scientific investigations are needed to improve the differential diagnosis between ADHD of a post-traumatic nature and ADHD of a neurobiological nature. Screening for childhood traumatic experiences and PTSD symptoms is warranted in ADHD children with an emphasis on detecting those with subthreshold PTSD symptoms. As the number of children with this diagnosis is increasing, the risk is that the number of children undergoing drug treatment, which is not always necessary, will increase in parallel. In addition, ADHD is the second most frequently diagnosed disorder among a population of sexually abused children with a primary diagnosis of PTSD [20].

The affective relationship as a process of affective attunement is recognized as a fundamental precursor of both the psychic and neurobiological development of the brain [21]. However, often this tuning may not take place for several reasons: due to maternal incapacity, due to unfavorable environments (think of situations of wars or other types of inconvenience) and due to illnesses of the child. Thus, among other things, emotion-related processes are compromised. The emotional regulation refers to the articulated affective, behavioral, and cognitive processes that coordinate the intensity, duration and expression of emotions linked to the internal and external stimuli. These emotion-related processes include emotions such anger and sadness, stress responses to circumstances that exceed an individual’s ability to cope, and moods such as depression and euphoria that could be present in both ADHD and PTSD [22]. Indeed, both these conditions share executive dysfunction, involving also behavioral regulation and emotional control cerebral network circuits related to amygdalae activation [23,24]. In the circumstance of adverse conditions which can contribute to the failure of these processes, the history of adoption can be configured as one of those aversive experiences that may compromise affective attunement and the shared attention. 

The evidence between ACEs and ADHD diagnosis remains, to date, controversial. To our knowledge, the possible explanations are the following: (1) there is a causal effect of ACEs on ADHD symptoms; (2) trauma-related symptoms are mistaken for ADHD symptomatology; (3) there is a bidirectional relationship between ADHD and ACEs. For adults and children who have experienced ACEs, the most recommended treatment is some type of trauma-informed care [24,25]. EMDR is an innovative treatment that has proven to be effective for a range of disorders, including PTSD, depression, anxiety, eating disorders and other illness [26].

To date, the literature had described only one case study of an adult patient with ADHD and history of infant trauma successfully treated with EMDR [27]. 

It is an integrative psychotherapy combining a variety of psychotherapeutic orientations, and its application is guided by an information-processing model [28]. Distress from childhood adversities can become encoded in state-specific form (or frozen in time) in its own neural network. Unless assimilated into adaptive memory networks, childhood distress retains the power that can lead to dysfunctional behaviors. The original perceptions of these painful memories, such as fear, can be triggered by current events that may lead to maladaptive responses and personality characteristics [29]. Eye Movement Desensitization and Reprocessing (EMDR) focuses on desensitization and reprocessing traumatic memories by facilitating dynamic linkages to adaptive memory networks [30]. EMDR uses standardized protocols and procedures that integrate affect, cognition, and response. Integrating adaptive memory networks into the network with the early distressing memory can result in building new neural connections and facilitate learning and adaptive storage of memories [31]. This process of integrating affect, cognition and response increases cerebral perfusion in the anterior cingulate cortex (ACC) [32] and dorsolateral prefrontal cortex [33] with activation of these areas. The dorsolateral prefrontal cortex is believed to be involved in memory, speech, and cognition and to belong to the neural circuitry of traumatic stress [34]. More specifically, the structure is involved in executive function and working memory. A recent study also suggests that the mechanism of therapeutic effectiveness in EMDR may be as follows: (a) emotional regulation by increased activity of the prefrontal lobe, (b) inhibition of overstimulation in the amygdalae by regulating the association cortex, (c) transformation of past traumatic memory, and (d) induction of functional balance between the limbic system and the prefrontal lobe. 

## 4. Conclusions 

EMDR [35] could be considered as a potentially useful adjunctive treatment for children and adolescents with ADHD suffering from traumatic childhood experiences. 

Recent neuroscience findings have demonstrated the role of the primary affective processes on memories consolidation through thalamic–hypothalamic pathways and additional studies should further investigated on subcortical neural correlates [36].

Future randomized clinical trials will be needed to confirm the efficacy of EMDR treatment on young ADHD patients with ACEs. 

## Figures and Tables

**Table 1 jpm-13-00200-t001:** Self-report questionnaires before EMDR therapy.

Self-Report Questionnaire		Scores
Mental functioning	** *PID-5* **	**Negative Affect = 1.694** **Detachment = 1.870** **Antagonism = 1.191** **Disinhibition = 1.358** Psychoticism *= 0.551* *Clinical cut-off > 1 (Max = 3)*
Ability to recognize and discriminate their emotional experiences	** *TAS-20* **	F1 = 23 F2 = 25 F3 =34 **TAS TOT: 82** *<51 = no alexithymia* *52–60 = border alexithymia* *>61 = alexithymia*
Traumatic and aversive childhood experiences	** *CTQ* **	Emotional Abuse (5–16+) = 4 Physical Abuse (5–13+) = 5 Sexual Abuse (5–13+) = 0 **Emotional Neglect (5–18+) = 10** Physical Neglect (5–13+) = 4 moderate traumatic experience Emotional neglect ≥ 10
	** *ACE* **	8 ACE does not give a clinical score but highlights exposure to traumatic experiences. Max score 10. There are at least 8 adverse childhood experiences in this case
Emotional regulation and the degree of dysregulation present	** *DERS* **	Non accept = 9 Goals = 23 Awareness = 18 Strategies = 26 Clarity = 18 Impulse = 8 **Total scores = 102** Significant score > 80 Scores are presented as a total score as well as a score for each of the 6 subscales. Higher scores suggest greater problems with emotion regulation.
	** *DES II* **	Depersonalization/Derealization Factor = 13 Absorption Factor = 25 Amnesia Factor = 28 **DES TOT = 42.86** Significant score >30
Symptoms and clinical severity	** *SCL-90 R* **	**Somatization = 1.67** **Obsessive-compulsive = 2.30** **Interpersonal sensibility = 3.11** **Depression = 2.54** **Anxiety = 3.30 Anger-hostility = 2.17** **Phobic-anxiety = 1.29** **Paranoid ideation = 3.33** **Psychoticism = 2.10** Significant score >1 up to 3 **GSI–General Symptomatic Index = 2.40**
**In bold are reported significant scores**		

**Table 2 jpm-13-00200-t002:** Neuropsychological assessments before and after EMDR therapy.

	Test	Scores before EMDR	Scores Post EMDR	Clinical Cut-Off
** *Memory* **	*Rey–Osterrieth complex figure Recall*	*22 (15.75)*	*26 (16.5)*	9.47
** *Deductive-logical reasoning* **	*Raven’s Colored Progressive Matrices*	*28 (25.8)*	*33 (30.8)*	18.96
** *sustained attention and set shifting* **	*Trail Making Test*	*A: 85”* *B: 135”*	*A: 67”* *B: 78”*	>93 >282
** *Executive functions* **	*Rey–Osterrieth complex figure Copy*	**29 (25.5)**	*35 (31.5)*	28.88
**In bold are reported significant scores**				

## Data Availability

Data is contained within the article.

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
