# Peer review of "Eye Movement Desensitization and Reprocessing (EMDR) as a Possible Evidence-Based Rehabilitation Treatment Option for a Patient with ADHD and History of Adverse Childhood Experiences: A Case Report Study"

_jpm, 2023, doi:10.3390/jpm13020200_

Round 1
Reviewer 1 Report
Dear Authors.
The idea of ​​the manuscript is of interest in the clinical field especially. However, I present some observations necessary to clarify:
In the abstract, in the second line, they use the term "challenging" around the therapeutic process. Although the challenge it means for a therapist is understandable, it is rather because the therapies in this context imply a degree of complexity given the diversity of expression of the symptoms for people with ADHD. Further on they say they express that "ADHD-related symptoms are important to prevent misdiagnosis", however, it is not the objective of the manuscript. Rather it is a projection to be discussed below, but not in the abstract.
In the introduction,
Regarding traumatic experiences and their impact on children, there should be some citation that explicitly corroborates it. For example, 10.1016/j.acap.2021.03.009.
On the other hand, without detracting from the cited report (1), they should refer to more recent WoS indexing findings. For example: for (1) 10.1016/j.chiabu.2019.104127, for (2) 10.1016/j.psychres.2018.08.097.
• If you talk about brain structures, you should refer to publications that support it. The APA is a secondary source. I suggest supporting it with primary and updated references. Likewise, what refers to the characteristics of trauma and its implications? The bibliographic foundation is insufficient to support affirmations of possible cortical alterations. Mentioning abnormalities of the prefrontal cortex requires fundamental neuroanatomical evidence. The functioning of the individual, or the behavior attributed to the neuropsychological basis, such as executive functions, can present a complex cortical correlate and is not only attributed to the frontal cortex. In this, there is more recent evidence than the one indicated. Regarding the above, an interesting paper reviews the symptomatic similarity between PTSD and ADHD: 10.1177/1087054716677818
• In the introduction, the expression of the differential diagnosis is missing. On the one hand, the diagnostic hypothesis is ADHD, differentially PTSD, how are they different? What are the shreds of evidence that should be taken into account to rule it out? Trauma as an independent variable is also complex to examine, in its acute or chronic genesis. This requires more explanation.
• In the description of the case, the normal cognitive profile obtained by the administration of the WISC IV stands out, with indices above 50% according to the standard (QIT value of 121 (verbal comprehension index (VCI) 124; reasoning index (PRI) 132; Processing Speed ​​Index (PSI) 115; Working Memory Index (WMI) 91). For a child with the situations of adversity described, these indices can be seen in a context of resilience that has allowed adapt so far. The decline in academic and social functioning requires more evidence, especially in its genesis. Is it possible to consider other antecedents that occurred at that age or earlier? Is this relevant to formulate the differential hypothesis of PTSD? Normally a person with ADHD presents low cognitive performance according to standards. It is important to review this point.
• The type of cognitive behavioral therapy is important, at least referring to its characteristics that allow for evaluating its relative success. This, plus the clinical evaluations motivated the application of the EMDR technique. The question here is why this technique or therapeutic strategy and not others? What part of the diagnosis motivates the decision to apply EMDR?
• On the other hand, the post-treatment evaluation is relatively equivalent to the pre-intervention cognitive evaluations. In it, they are all non-verbal. Conceptual management can have a critical impact on ADHD symptomatology. Are the neuropsychological tests chosen related to the results obtained with EMDR?
• It is important to describe how this EMDR strategy was carried out by the research team. These characteristics are essential to establish its effectiveness in the indicated diagnosis.
• In paragraph 3 of the discussion, when referring to affective states, they resort to text citations that refer to it in general terms. These affective phenomena in ADHD or PTSD present interesting neurophysiological and neuropsychological characteristics that refer precisely to these diagnoses, rather than a generic description. It is suggested that they could base these aspects in a precise way that contributes to the understanding of ADHD and after the EMDR technique that could contribute to its stabilization.
In the final paragraph of the discussion, they refer to EMDR. Referring to this expression "more innovative approaches to treat the symptoms of post-traumatic stress disorder" there should be some citation that reports results in this regard to be able to demonstrate said affirmation. In the most up-to-date investigations, there is more recent evidence of the neurocortical impact, especially in some investigations that are in the psychotherapeutic context for this type of disorder, for example 10.22365/jpsych.2020.312.162
Finally, in conclusion, the background is not as conclusive as expressed above. We cannot say from a case or a reference that the application of the EMDR approach has been successful. Precisely, they previously mentioned that it requires further investigation. Likewise, with recent studies that contribute to affective neurocognitive functioning and its link with executive processes, they must be updated, in light of recent evidence in this regard. It may be an interesting approach, but more evidence is needed in this regard. For example, in this case, study, what was it that changed psychologically and behaviorally in the evaluated person with this treatment? Is it possible to differentiate it from other types of causes?
Author Response
The idea of ​​the manuscript is of interest in the clinical field especially. However, I present some observations necessary to clarify:
In the abstract, in the second line, they use the term "challenging" around the therapeutic process. Although the challenge it means for a therapist is understandable, it is rather because the therapies in this context imply a degree of complexity given the diversity of expression of the symptoms for people with ADHD. Further on they say they express that "ADHD-related symptoms are important to prevent misdiagnosis", however, it is not the objective of the manuscript. Rather it is a projection to be discussed below, but not in the abstract.
We thank the reviewer for the observations that have been considered. The test in the abstract has been edited
In the introduction,
Regarding traumatic experiences and their impact on children, there should be some citation that explicitly corroborates it. For example, 10.1016/j.acap.2021.03.009.
This reference has been added
On the other hand, without detracting from the cited report (1), they should refer to more recent WoS indexing findings. For example: for (1) 10.1016/j.chiabu.2019.104127, for (2) 10.1016/j.psychres.2018.08.097.
These references have been added
- If you talk about brain structures, you should refer to publications that support it. The APA is a secondary source. I suggest supporting it with primary and updated references. Likewise, what refers to the characteristics of trauma and its implications? The bibliographic foundation is insufficient to support affirmations of possible cortical alterations. Mentioning abnormalities of the prefrontal cortex requires fundamental neuroanatomical evidence. The functioning of the individual, or the behavior attributed to the neuropsychological basis, such as executive functions, can present a complex cortical correlate and is not only attributed to the frontal cortex. In this, there is more recent evidence than the one indicated. Regarding the above, an interesting paper reviews the symptomatic similarity between PTSD and ADHD: 10.1177/1087054716677818
We thank the reviewer and this reference has been added
- In the introduction, the expression of the differential diagnosis is missing. On the one hand, the diagnostic hypothesis is ADHD, differentially PTSD, how are they different? What are the shreds of evidence that should be taken into account to rule it out? Trauma as an independent variable is also complex to examine, in its acute or chronic genesis. This requires more explanation.
We thank the reviewer and this difference has been added
- In the description of the case, the normal cognitive profile obtained by the administration of the WISC IV stands out, with indices above 50% according to the standard (QIT value of 121 (verbal comprehension index (VCI) 124; reasoning index (PRI) 132; Processing Speed ​​Index (PSI) 115; Working Memory Index (WMI) 91). For a child with the situations of adversity described, these indices can be seen in a context of resilience that has allowed adapt so far. The decline in academic and social functioning requires more evidence, especially in its genesis. Is it possible to consider other antecedents that occurred at that age or earlier? Is this relevant to formulate the differential hypothesis of PTSD? Normally a person with ADHD presents low cognitive performance according to standards. It is important to review this point.
We thank the reviewer for the comment. Unfortunately this is the first assessment made in our Hospital and we have no news about previous neuropsychological assessments. We added another WISC IV assessment performed at that age of 15 years (QIT value of 107, VCI 104, PRI 112, PSI 103, WMI 85) associated to severely learning difficulties. It has been added in the test.
- The type of cognitive behavioral therapy is important, at least referring to its characteristics that allow for evaluating its relative success. This, plus the clinical evaluations motivated the application of the EMDR technique. The question here is why this technique or therapeutic strategy and not others? What part of the diagnosis motivates the decision to apply EMDR?
We thank the reviewer for the comment The patient refused to continue CBT therapy. EMDR therapy has been considered due to the traumatic experience and the collaborative efforts available in our Hospital with the territorial Presidium Prevention of Early Interventions Mental Health (PIPSM). It has been added in the test.
- On the other hand, the post-treatment evaluation is relatively equivalent to the pre-intervention cognitive evaluations. In it, they are all non-verbal. Conceptual management can have a critical impact on ADHD symptomatology. Are the neuropsychological tests chosen related to the results obtained with EMDR?
We thank the reviewer for the comment. The neuropsychological tests have part of clinical assessment performed before starting EMDR therapy and 9 months later. We reported table 2 with all the assessments made before and after 9 months of EMDR treatment.
- It is important to describe how this EMDR strategy was carried out by the research team. These characteristics are essential to establish its effectiveness in the indicated diagnosis.
As it has been reported in the test “The EMDR procedure were explain and have obtained the consent to the treatment. After due to the consideration of the traumatic memories related to his mothers’ death (when his mother died, he was 3 years old and he remembers only a song sent by a radio: a very sad song), he was treated on this target with EMDR for two 65 minutes sessions”.
How it has been reported in the Table 2 the boy improved in executive functions.
- In paragraph 3 of the discussion, when referring to affective states, they resort to text citations that refer to it in general terms. These affective phenomena in ADHD or PTSD present interesting neurophysiological and neuropsychological characteristics that refer precisely to these diagnoses, rather than a generic description. It is suggested that they could base these aspects in a precise way that contributes to the understanding of ADHD and after the EMDR technique that could contribute to its stabilization.
We thank the reviewer for the observation that has been considered and added in the test with reference 23.
In the final paragraph of the discussion, they refer to EMDR. Referring to this expression "more innovative approaches to treat the symptoms of post-traumatic stress disorder" there should be some citation that reports results in this regard to be able to demonstrate said affirmation. In the most up-to-date investigations, there is more recent evidence of the neurocortical impact, especially in some investigations that are in the psychotherapeutic context for this type of disorder, for example 10.22365/jpsych.2020.312.162
We thank the reviewer for the observation. We have added the reference suggested reporting a study conducted in an adult patient [28]..
Finally, in conclusion, the background is not as conclusive as expressed above. We cannot say from a case or a reference that the application of the EMDR approach has been successful. Precisely, they previously mentioned that it requires further investigation. Likewise, with recent studies that contribute to affective neurocognitive functioning and its link with executive processes, they must be updated, in light of recent evidence in this regard. It may be an interesting approach, but more evidence is needed in this regard. For example, in this case, study, what was it that changed psychologically and behaviorally in the evaluated person with this treatment? Is it possible to differentiate it from other types of causes?
We thanks the reviewer for the observation. We added one case study of an adult patient with ADHD and history of infant trauma successfully treated with EMDR [28]. In our case after EMDR treatment both affective and executive functions improved
Reviewer 2 Report
This paper is a case report of the successful use of eye movement desensitization and reprocessing (EMDR) in a child with attention-deficit/hyperactivity disorder (ADHD) and a history of childhood trauma.
The report presented is innovative; there is only one prior case report on a related topic that I am aware of, and this involved an adult patient (Broad and Wheeler, 2006). However, the current report could be improved by corrections or clarifications in the following areas:
1. This paper has been submitted to a journal section focusing on the interface between clinical practice and underlying molecular biology / physiology. Therefore, the Introduction and Discussion could be improved by incorporating recent literature on the neurobiology of adverse childhood experiences (ACEs) in children with ADHD and the implications of this research for treatment.
2. The discussion of the links between ADHD and trauma is inconclusive. Are the authors stating that trauma is a risk factor for ADHD (possible), that ADHD is a risk factor for trauma (also possible) or both? This should be discussed with greater precision (e.g., are there longitudinal studies of subsequent ADHD in children exposed to ACEs at a very young age? are there studies of gene-environment interaction or correlation?)
3. The case history would benefit from some additional information:
a. Was there any information on ACEs faced by the child prior to the age of 3, or during his stay in the orphanage between the ages of 3 and 9?
b. What was the exact nature of the "substance abuse disorder" in the child's biological mother? Was the child exposed to the toxic effects of any substance in utero?
c. What was the quality of the child's attachment to his adoptive parents? Did he experience any ACEs after adoption (e.g., bullying at school?)
d. What were the child's developmental milestones and scholastic skills? Was there delay or deviance in any specific domain? Was there any general or specific learning disability?
e. Why did the child require hospitalization for ADHD? This disorder is usually treated on an out-patient basis unless there are comorbidities or severe behavioral problems causing a risk of harm to the child or others.
f. Did the child fulfill criteria for PTSD given his past history? If this was not assessed, why?
g. Was the "severe social and academic impairment" noted on p.2 due to ADHD, due to the sequelae of trauma, or due to another specific developmental disorder?
h. What was the indication for the use of aripiprazole in this child? This drug is currently not approved either for ADHD or for disruptive behavior disorders; risperidone would be a more logical choice and is better supported by the evidence.
i. Did the child fulfill criteria for any other comorbid diagnosis, specifically a mood disorder (depression or bipolar) or a disruptive behavior disorder (conduct disorder or oppositional-defiant disorder)?
4. The authors have presented symptom severity scores prior to EMDR and neuropsychological test scores after EMDR. It would be more logical to present pre- and post-treatment symptom and cognitive test scores in the same table, and then examine which parameters had changed post-EMDR.
5. Attachment disorders can mimic ADHD; was this child evaluated for a reactive attachment disorder given his traumatic early history?
6. The Discussion and Conclusions should be better organized and should cover the issues mentioned in points 1, 2 and 5 above. The earlier case report from 2006 can also be cited and discussed in this context.
7. Language editing is required to address sentence fragments ("as defined by DSMV.referenza") and spelling errors ("talamo-cortical" should be "thalamo-cortical").
8. It is not clear why the authors refer to the Declaration of Helsinki and mention a "study" - this is a single case report. What should be documented instead is informed consent from the parent/ guardian and assent from the child.
Author Response
This paper is a case report of the successful use of eye movement desensitization and reprocessing (EMDR) in a child with attention-deficit/hyperactivity disorder (ADHD) and a history of childhood trauma.
The report presented is innovative; there is only one prior case report on a related topic that I am aware of, and this involved an adult patient (Broad and Wheeler, 2006). However, the current report could be improved by corrections or clarifications in the following areas:
We thank the reviewer for the comment. We added this reference in the discussion.
- This paper has been submitted to a journal section focusing on the interface between clinical practice and underlying molecular biology / physiology. Therefore, the Introduction and Discussion could be improved by incorporating recent literature on the neurobiology of adverse childhood experiences (ACEs) in children with ADHD and the implications of this research for treatment.
We thank the reviewer for the observations. The introduction and the discussion have been amended.
- The discussion of the links between ADHD and trauma is inconclusive. Are the authors stating that trauma is a risk factor for ADHD (possible), that ADHD is a risk factor for trauma (also possible) or both? This should be discussed with greater precision (e.g., are there longitudinal studies of subsequent ADHD in children exposed to ACEs at a very young age? are there studies of gene-environment interaction or correlation?)
We thank the reviewer for the observations. This has been added in the discussion.
- The case history would benefit from some additional information:
- Was there any information on ACEs faced by the child prior to the age of 3, or during his stay in the orphanage between the ages of 3 and 9?
It was also reported that the child assisted to the death of 2 sisters by a car accident when he was young than 9 years. When his mother died by an overdose, he was 3 years old and he remembered only a song sent by a radio: a very sad song. These informations have been amended in the test.
- What was the exact nature of the "substance abuse disorder" in the child's biological mother? Was the child exposed to the toxic effects of any substance in utero?
His mother died by an overdose (narcotic); there are no news about the exposition during utero.
- What was the quality of the child's attachment to his adoptive parents? Did he experience any ACEs after adoption (e.g., bullying at school?)
He experienced ACEs due to the removal of the sister's house to which he was very attached by the adoptive parents due to psychiatric problems that arose in the sister. This information has been added in the test.
- What were the child's developmental milestones and scholastic skills? Was there delay or deviance in any specific domain? Was there any general or specific learning disability?
The child's developmental milestones were reported as normal. He developped specific learning difficulties. These informations have been added in the test
- Why did the child require hospitalization for ADHD? This disorder is usually treated on an out-patient basis unless there are comorbidities or severe behavioral problems causing a risk of harm to the child or others.
We correct the test. He was treated on an outpatient basis at our hospital. This information has been amended.
- Did the child fulfill criteria for PTSD given his past history? If this was not assessed, why?
We thank the reviewer for the comment. A mild form of PTSD diagnosis has been considered and it has been added in the test.
- Was the "severe social and academic impairment" noted on p.2 due to ADHD, due to the sequelae of trauma, or due to another specific developmental disorder?
The academic impairment was imputable to ADHD symptoms. Specifically, his academic impairment was attributed to lack of attention and to specific learning difficulties. His social impairment was probably imputable to irritability, anger and social withdrawal.
- What was the indication for the use of aripiprazole in this child? This drug is currently not approved either for ADHD or for disruptive behavior disorders; risperidone would be a more logical choice and is better supported by the evidence.
Aripripazole has been chosen to the less side effects than risperidone and due to the efficacy reported in the literature. Tthere is growing evidence about the effectiveness of the aripiprazole/methylphenidate combination in patient with ADHD presenting irritability. This is a reference supporting this hypothesis: Pan PY, Fu AT, Yeh CB. Aripiprazole/Methylphenidate Combination in Children and Adolescents with Disruptive Mood Dysregulation Disorder and Attention-Deficit/Hyperactivity Disorder: An Open-Label Study. J Child Adolesc Psychopharmacol. 2018 Dec;28(10):682-689. doi: 10.1089/cap.2018.0068. Epub 2018 Aug 27. PMID: 30148656.
- Did the child fulfill criteria for any other comorbid diagnosis, specifically a mood disorder (depression or bipolar) or a disruptive behavior disorder (conduct disorder or oppositional-defiant disorder)?
He presented symptoms including anger, irritability, loss of control and affective instability oppositional-defiant attittudes compatible with depressive like symptoms without fulfilling criteria for mood disorders.
- The authors have presented symptom severity scores prior to EMDR and neuropsychological test scores after EMDR. It would be more logical to present pre- and post-treatment symptom and cognitive test scores in the same table, and then examine which parameters had changed post-EMDR.
We thank the reviewer for the comment. This has been added in the table 2
- Attachment disorders can mimic ADHD; was this child evaluated for a reactive attachment disorder given his traumatic early history?
We thank the reviewer for the comment. An avoidant attachment disorders was made during EMDR therapy. Even if the attachment disorder can mimic ADHD in this case the behavioural and neuropsychological findings were compatible with ADHD according to DSM V criteria, but the severity was influenced by the attachment disorder. This has been added in the test.
- The Discussion and Conclusions should be better organized and should cover the issues mentioned in points 1, 2 and 5 above. The earlier case report from 2006 can also be cited and discussed in this context.
We thank the reviewer for the comment. These have been amended.
- Language editing is required to address sentence fragments ("as defined by DSMV.referenza") and spelling errors ("talamo-cortical" should be "thalamo-cortical").
We thank the reviewer for the comment. These have been amended.
- It is not clear why the authors refer to the Declaration of Helsinki and mention a "study" - this is a single case report. What should be documented instead is informed consent from the parent/ guardian and assent from the child.
These informations have been made according to the submission guidelines of the journal
Reviewer 3 Report
JPM-2033137 v1 – Peer Review – England, 28th of November 2022.
The case report by Guidetti et alia is presenting evidence that EMDR was efficient to rehabilitate a patient suffering from ADHD and adverse childhood experiences.
At present, the article needs several alterations before publication. Indeed, the EMDR procedure is not described. Furthermore, the title needs to be interpreted with caution: the current study has observed efficacy of the EMDR treatment in a single patient, which cannot be extrapolated to “patients” (as written in the title). Only a RCT can prove/disprove such a result in a large population of “patients”.
The authors should revise their manuscript accordingly (major revision), which should not take long.
Can the authors please focus on the following:
-Please improve the grammar, throughout.
-Please adjust the title to reflect the comment above.
-Please give additional details about the EMDR treatment (including instruments and protocols).
-In the methodology, please also include the normal/healthy ranges of scores for all parameters listed in Table 1 and Table 2.
-Please insert additional details in the legends of the two tables (what does “cut-off” refer to?). Although this might be clear for clinicians, this should be explained in details for readers that are not clinicians.
I am looking forward to reviewing the revised manuscript.
Author Response
The case report by Guidetti et alia is presenting evidence that EMDR was efficient to rehabilitate a patient suffering from ADHD and adverse childhood experiences.
At present, the article needs several alterations before publication. Indeed, the EMDR procedure is not described. Furthermore, the title needs to be interpreted with caution: the current study has observed efficacy of the EMDR treatment in a single patient, which cannot be extrapolated to “patients” (as written in the title). Only a RCT can prove/disprove such a result in a large population of “patients”.
We thank the reviewer for the comment. The title has been edited. The need of RCT has been added in the conclusion.
The authors should revise their manuscript accordingly (major revision), which should not take long.
Can the authors please focus on the following:
-Please improve the grammar, throughout.
English has been revised.
-Please adjust the title to reflect the comment above.
We thank the reviewer for the comment. This has been added
-Please give additional details about the EMDR treatment (including instruments and protocols).
We thank the reviewer for the comment. The EMDR procedure were explain and have obtained the consent to the treatment. After due to the consideration of the traumatic memories related to his mothers’ death (when his mother died, he was 3 years old and he remembers only a song sent by a radio: a very sad song), he was treated on this target with EMDR for two 65 minutes sessions.
-In the methodology, please also include the normal/healthy ranges of scores for all parameters listed in Table 1 and Table 2.
We thank the reviewer for the comment. This has been added
-Please insert additional details in the legends of the two tables (what does “cut-off” refer to?). Although this might be clear for clinicians, this should be explained in details for readers that are not clinicians.
We thank the reviewer for the comment. This has been added
Round 2
Reviewer 1 Report
Dear Authors.
The manuscript has considered all the observations presented in the first review. In this, evidence has been presented in the same manuscript of the changes and clarifications proposed, which improves its presentation. Therefore, there are no objections to its approval and subsequent publication.
Just change the name of the DSM 5 manual, which is mentioned with Roman numerals. It should be presented with Arabic numbers.
Author Response
The manuscript has considered all the observations presented in the first review. In this, evidence has been presented in the same manuscript of the changes and clarifications proposed, which improves its presentation. Therefore, there are no objections to its approval and subsequent publication.
We thank the reviewer for the comment.
Just change the name of the DSM 5 manual, which is mentioned with Roman numerals. It should be presented with Arabic numbers.
This has been amended
Reviewer 2 Report
The revisions made by the authors are satisfactory. I have no further major corrections or changes to suggest from a scientific standpoint; however, the paper still requires language editing to address numerous errors in grammar and sentence structure (e.g. "EMDR treatment could be efficacy on childhood traumatic experiences", "The emotional dysregulation symptoms have subsided with attentive abilities growth and him inclusion in a therapeutic treatment group", "He started cognitive behavioral psychotherapy intervention (CBT) once a week with a good efficacy").
Author Response
The revisions made by the authors are satisfactory. I have no further major corrections or changes to suggest from a scientific standpoint; however, the paper still requires language editing to address numerous errors in grammar and sentence structure (e.g. "EMDR treatment could be efficacy on childhood traumatic experiences", "The emotional dysregulation symptoms have subsided with attentive abilities growth and him inclusion in a therapeutic treatment group", "He started cognitive behavioral psychotherapy intervention (CBT) once a week with a good efficacy").
We thank the reviewer for the comment. The English language has been edited
Reviewer 3 Report
England, 27th of December 2022.
Thank you for sending me the revised manuscript (JPM-2033137-v2).
Whilst some of my concerns were addressed, unfortunately, the methodology of EMDR remains unexplained in the manuscript (as higlighted previously, v1):
-Please give additional details about the EMDR treatment (including instruments and protocols).
Which is the exact protocol that was used? For reference, authors might want to higlight their protocol/methodology from the following two references:
a) Shapiro, F. (1989). Efficacy of the eye movement desensitization procedure in the treatment of traumatic memories. J. Trauma Stress 2, 199–223. doi: 10.1002/jts.2490020207
b)
Shapiro, F. (1994). Eye Movement Desensitization and Reprocessing: Basic Principles, Protocols and Procedures. New York, NY: Guilford Press.
Finally, the grammar needs to be improved in the newly-inserted text sections (red).
Author Response
Thank you for sending me the revised manuscript (JPM-2033137-v2).
Whilst some of my concerns were addressed, unfortunately, the methodology of EMDR remains unexplained in the manuscript (as higlighted previously, v1):
-Please give additional details about the EMDR treatment (including instruments and protocols).
Which is the exact protocol that was used? For reference, authors might want to higlight their protocol/methodology from the following two references:
a)Shapiro, F. (1989). Efficacy of the eye movement desensitization procedure in the treatment of traumatic memories. J. Trauma Stress 2, 199–223. doi: 10.1002/jts.2490020207
b)
Shapiro, F. (1994). Eye Movement Desensitization and Reprocessing: Basic Principles, Protocols and Procedures. New York, NY: Guilford Press.
Finally, the grammar needs to be improved in the newly-inserted text sections (red).
We thank the reviewer for the comment.
We have initially chosen not to describe the EMDR therapy specifically because it is a well-known therapeutic practice.
EMDR therapy is very common around the world. In the United States, the Department of Veterans Affairs and Department of Defense list EMDR as a “best practice” in treating veterans experiencing PTSD. Research on EMDR includes dozens of clinical trials, research studies and academic papers. It has official approval from the World Health Organization (WHO) and government organizations and agencies in the United Kingdom, Australia and Germany, among others
Only licensed therapists can practice EMDR therapy after certified training by EMDR Institutions, and in our case by EMDR Italia. Dr. Patrizia Brogna is a Level 2 EMDR praticioner eith extensive esperience in treatment of patients with Post-Traumatic Stress Disorders (PTSD).
As requested by the reviewer we report the protocol used adding this in the test: “We used the standard Protocol of EMDR therapy on past memories. EMDR therapy consists of eight phases. These phases occur over multiple sessions, with one session sometimes using parts of several phases. An example of this would be how phases 1 and 2 typically happen only in early sessions, while phases 3 through 8 are part of multiple sessions later. For a single disturbing event or memory, it usually takes between three and six sessions. More complex or longer-term traumas may take eight to 12 sessions (or sometimes more). Sessions usually last between an hour and 90 minute”s.
The eight phases of EMDR have been reported in table S1 as Supplementary file S1
Table S1: The eight phases of EMDR therapy
1 |
Patient history and information gathering. This part of the process involves your healthcare provider gathering information about you and your past. This helps them determine if EMDR is likely to help you. It also includes asking about upsetting or disturbing events and memories that you want your therapy to focus on, as well as your goals for this therapy. |
2 |
Preparation and education. During this phase, your healthcare provider will talk to you about what will happen during EMDR sessions and what you can expect. They’ll also talk to you about things to focus on to help you feel more stable and safer during sessions. They’ll provide you with tools to help you manage your emotions. |
3 |
Assessment. This part of the process is where your healthcare provider helps you identify themes and specific memories that you may want to work on during reprocessing. They’ll help you identify both negative beliefs about how the trauma has made you feel, as well as positive beliefs that you would like to believe about yourself going forward. |
4 |
Desensitization and reprocessing. During this phase, your healthcare provider activates your memory by helping you identify one or more specific negative images, thoughts, feelings and body sensations. Throughout the reprocessing, they’ll help you notice how you feel and any new thoughts or insight you have about what you’re experiencing. |
5 |
Installation. During this phase, your healthcare provider will have you focus on the positive belief you want to build in as you process a memory. This positive belief can be what you said in phase 3 or something new you think of during phase 4. |
6 |
Body scan. Your healthcare provider will have you focus on how you feel in your body, especially any of the symptoms you feel when you think about or experience the negative memory. This phase helps identify your progress through EMDR therapy overall. As you go through sessions, your symptoms should decrease until you don’t have any (or as close to none as possible). Once your symptoms are gone, your reprocessing is complete. |
7 |
Closure and stabilization. This phase forms a bridge between later sessions. During this phase, your healthcare provider will talk to you about what you should expect between sessions. They’ll also talk to you about how to stabilize yourself, especially if you have negative thoughts or feelings during the time between sessions. They won’t end a session until you feel calmer and safe. They might also ask you to write down any new thoughts you have about the disturbing event(s), so you can bring them up at your next session. |
8 |
Reevaluation and continuing care. The final phase of EMDR therapy involves your healthcare provider going over your progress and how you’re doing now. This can help determine if you need additional sessions or how to adjust your goals and expectations for your therapy. They’ll also help you explore what you might experience in the future — how you would like to handle things at that time, knowing what you know now, about yourself and your past trauma. |